# Demography and perturbation analyses of the coffee berry borer *Hypothenemus hampei* (Coleoptera: Curculionidae): Implications for management

**Yobana A. Mariño** *, **Paul Bayman, Alberto M. Sabat**

Department of Biology, University of Puerto Rico – Río Piedras, San Juan, Puerto Rico, United States of America

* yandreamarinoc@gmail.com, yobana.marino@upr.edu

**Data Availability Statement:** All relevant data are within the paper and its Supporting information files.

## Abstract

The coffee berry borer (CBB) *Hypothenemus hampei* Ferrari is the most serious pest of coffee worldwide. Management of the CBB is extremely difficult because its entire life cycle occurs inside the fruit, where it is well protected. Knowing which life stages contribute most to population growth, would shed light on the population dynamics of this pest and help to improve CBB management programs. Two staged-classified matrices were constructed for CBB populations reared in the lab on artificial diets and CBB populations from artificial infestations in the field. Matrices were used to determine demographic parameters, to conduct elasticity analyses, and to perform prospective perturbation analysis. Higher values of the intrinsic rate of natural increase ($r_m$) and population growth rate (λ): were observed for CBB populations growing in the lab than in the field ($r_m$: 0.058, λ: 1.74 lab; $r_m$: 0.053, λ: 1.32 field). Sensitivity values for both CBB populations were highest for the transitions from larva to pupa ($G_2$: 0.316 lab, 0.352 field), transition from pupa to juvenile ($G_3$: 0.345 lab, 0.515 field) and survival of adult females ($P_5$: 0.324 lab, 0.389 field); these three vital rates can be important targets for CBB management. Prospective perturbation analyses indicated that an effective management for the CBB should consider multiple developmental stages; perturbations of >90% for each transition are necessary to reduce λ to <1. However, when the three vital rates with highest sensitivity are impacted at the same time, the percentage of perturbation is reduced to 25% for each transition; with these reductions in survival of larvae, pupae and adult females the value of λ was reduced from 1.32 to 0.96. Management programs for CBB should be focused on the use of biological and cultural measures that are known to affect these three important targets.

## Introduction

Coffee, *Coffea arabica L.* (Gentianales: Rubiceae) is one of the most valuable commodities in the global market, and its exportation is a key source of income for many developing countries [1, 2]. Coffee is grown in more than 70 countries in the humid tropics [3]. In Latin America, coffee is produced from Mexico and the Caribbean to Brazil [4, 5].

**Funding:** USDA Specific Cooperative Agreements 58-1245-4-082 and 58-1245-4-083. Puerto Rico Science and Technology Research Trust, grant ARG 2022-00042. The funders had no role in study design, data collection and analysis, decision to publish, or preparation of the manuscript.

**Competing interests:** The authors have declared that no competing interests exist.

The coffee berry borer *Hypothenemus hampei* (CBB) (Ferrari) (Coleoptera: Curculionidae: Scolytinae) is the most serious pest of coffee worldwide [3, 6, 7]. The coffee berry borer is endemic to central Africa but has been disseminated to most coffee-producing countries around the world [8, 9]. Adult females bore into the coffee fruit; when they reach the seed's endosperm, they make galleries and lay eggs. The entire life cycle is completed inside the fruit; only fertilized females abandon the fruit in search of new fruits to infest [7]. Damage caused by the CBB is due to feeding activity of females and their progeny in the coffee seeds, which reduce the quality and yield of the marketable product and induce the premature fall of green fruits. Additionally, holes made by the CBB can facilitate the entry of pathogens which cause internal fruit rot [3, 6, 10].

Management of the CBB is extremely difficult; chemical and biological control measures are only effective during the short time when the adult females are out searching for new fruits to infest. One of the most important biological insecticides used for management of the CBB is the fungus *Beauveria bassiana* (Balsamo) Vulleimen [11, 12]. In many countries, *B. bassiana* is commonly applied using the commercial products Mycotrol® and Botanigard®.

## The CBB life cycle

The life cycle of the CBB consists of egg, larva, pupa, juvenile and adult. Females have two larval instars, while males have only one [13, 14]. The body of the larvae is white and the head is brown, they are apodous, eucephalous and curved [15]. The number of instars is defined according to the width of the head capsule: the head capsule of instar I or larva I is 0.20–0.22 mm versus 0.28–0.32 mm for larva II [13]. Other differences between these two instars are body size and activity. Larva I is smaller (0.63–1.3 mm long) and moves less. Larva II is bigger (1.7–2.0 mm long), and is more active and voracious [13, 14].

Before becoming prepupae, larvae stop feeding and moving [13, 16]. Prepupae are bigger (2.2 mm long), less curved, and the thorax region is whiter and wider than larvae II [13]. But the prepupa is not always acknowledged as a stage of development of the CBB; some authors consider it as a stage distinct from a larva [14] and others consider it as a second larval instar [16]. Gómez et al. [13] observed that larvae molted only once before developing into pupae, and also that the width of the head capsule of prepupae overlapped with that of larvae II (0.28–0.30 mm), they considered it as part of the second larval instar. For our analysis we considered the prepupae as part of the larval instar II as stated by Gómez et al. [13].

Pupae are type exarate: they are white but their color changes as they develop [14–16], turning light brown as wings, legs and antennae begin to form. Size differs between the sexes: female pupae are bigger, 1.89 mm long, compared with 1.22 mm long for males [15, 16]. Young adults or juveniles are light brown, turning dark after two to five days, which indicates that they are sexually mature; the difference in size is maintained, as females are bigger than males (1.64–1.85 mm vs. 1.08–1.12 mm) [15, 16].

## Demography as a tool to study pests

Demographic perturbation models are an important tool to understand and manage populations, specifically those models that focus on sensitivity and elasticity analyses [17, 18]. These types of analyses can be used in two ways: to protect endangered species or to reduce populations of invasive pests and weeds [18]. However, most of these analyses have been focused on conservation of endangered populations and less on control of pests.

Sensitivity analysis estimates the effect of changes in vital rates (e.g., survival, growth, fecundity) on the population growth rate ($\lambda$). Elasticity is the proportional sensitivity and determines the proportional change in $\lambda$ caused by a proportional change in a vital rate [17, 19, 20].

This type of analysis is called prospective analysis because it predicts the results of vital rate perturbations before they happen. Prospective analysis can be used to identify potential management targets (vital rates with higher scores of sensitivities or elasticities), that can be modified to produce large changes in the value of λ [17].

Demographic studies or life table studies of the CBB have been done under both laboratory and field conditions [21–25]. These studies estimated important demographic parameters including: population growth rate (λ), net reproductive rate ($R_0$), intrinsic rate of natural increase ($r_m$) and mean generation time (T). Some of these studies show that demographic parameters are affected by temperature, type of diet, coffee variety and fruit age [22–24, 26]. However, none of these prior studies used demographic data to conduct perturbation analysis, specifically prospective analysis. Knowledge about population dynamics and sensitivity of vital rates with potential to be perturbed is essential for informed management decisions.

The goal of the present study was to perform artificial infestations of the CBB under field conditions and to rear the CBB under lab conditions, in order to estimate demographic transitions in its life cycle and to conduct perturbation analysis. The specific objectives were: (i) To compare demographic parameters of populations raised in the lab on an artificial diet (Cenibroca) [27] with those from artificial infestations in the field. (ii) To identify the contribution of vital rates to differences in performance between populations raised in the lab vs. artificial infestation in the field. (iii) To compare the behavior in the field of females raised in the lab (artificial infestation) vs. those of naturally infested coffee fruits (natural infestation). (iv) To identify the vital rates most sensitive to perturbations, and (vi) to conduct prospective analysis with those vital rates than can be modified and used as important targets for the management of the CBB.

## Materials and Methods

### Insect rearing

Coffee berry borers (CBB) were reared on the artificial diet Cenibroca [27] as previously described [25]. CBBs were maintained in a growth chamber (Model 818, Thermo Scientific, Dubuque, IA) set at 25˚C (±1˚C), 80–96% relative humidity (RH) and complete darkness. The colony was established years before the experiments. Founder females were obtained from infested fruits of *Coffea arabica* L. cv. Catuai from a farm in Adjuntas, Puerto Rico (18˚10'42" N, 66˚44'36" W). The colony is maintained in the Biology Department of the University of Puerto Rico, Río Piedras.

### Laboratory evaluations

For laboratory evaluations, 100 eggs from $F_6$ (6th generation) of the lab colony were transferred to tubes with fresh Cenibroca diet, one per tube. Eggs were evaluated daily for 50 days to

**Populations of CBB evaluated**:

**CBB raised in the lab**: individuals reared in the lab on the artificial diet Cenibroca.

**Artificial infestation**: coffee fruits in field were infested by CBB females reared in the lab on the artificial diet Cenibroca.

**Natural infestation**: Coffee fruits infested by CBB females growing under natural conditions.

determine the transition to the next developmental stages, survival in each stage and developmental time from egg to adult. To determine survival and fecundity of adult females, 100 females approximately ten days old were transferred individually to new tubes with diet and were evaluated for 90 days. Monthly each female was removed from the diet and survival, oviposition and total population were determined. CBB individuals per tube (eggs, larvae, pupae, juveniles) were counted, and original alive adult females were transferred again to new tubes with diet.

## Artificial infestation in the field

Three infestations were made between August to September 2018; at this time of year most fruits are full-sized and are approximately 80 days old [28]. Infestations were carried out at the UPR Agricultural Experimental Station in Adjuntas, Puerto Rico (18˚ 10' 12" N, 66˚47'43" W, altitude 434 m.a.s.l.) on *Coffea arabica* L. cv. Limaní growing under shade of *Gliricidia sepium* (mother-of-cocoa, Fabaceae: Faboideae).

For each infestation ten coffee trees were selected; in each tree five to ten branches with >30 green, full-sized fruits were tagged; fruits that showed perforations made by the CBB were removed. Approximately 90 branches were artificially infested. CBBs for artificial infestation were obtained from the colony maintained on the artificial diet Cenibroca described above.

On the day of infestation, CBB females were surface-disinfected with 1% sodium hypochlorite and 0.002% Tween 80, and rinsed twice with sterile distilled water (dH₂O) (1 min each). Then groups of fifty females were transferred to petri dishes (6 cm diameter) lined with a sterile moist filter paper and sealed with paraffin. Once in the field the CBBs were put into clear plastic bags (35 cm x 11 cm), used as entomological sleeves to cover the experimental branches. Bags were tied to the branches and removed two days later. Perforated fruits were marked with nail polish; a different color was used for each infestation.

Fifty infested berries were collected weekly from 7 to 56 days after infestation. Fruits were transported to the laboratory for dissection; all CBB stages inside the fruits were counted (eggs, larvae I, larvae II, pupae, juveniles, and adults) and oviposition, survival, presence and position of adult females was determined. Position of colonizing females was classified as follows: A) fruit is perforated but the colonizing female is not present; B) fruit is perforated but the female has not reached the endosperm; C) the female has reached the endosperm but there is no reproduction, and D) the female has made her gallery in the endosperm, and immature stages are found [29].

No chemical or biological control for the pest was applied in this plot during the experiment to ensure that mortality or fruit abandonment were due to natural causes, such as natural infection by fungi or bacteria or coffee fruit rot [10]. CBB females were noted as killed by the entomopathogenic fungus *B. bassiana* when a white mycelium typical of *B. bassiana* was observed growing on dead colonizing females. Percent of internal fruit rot was evaluated following the scale described by Serrato-Diaz [10].

## Natural infestation

Fifty naturally field-infested fruits were collected weekly from August to December from the same plot where artificial infestations were performed. In the laboratory the same variables mentioned above were evaluated: total number of CBB individuals per fruit, survival, presence and position of colonizing females, number of adult females killed by the fungus *B. bassiana* and percentages of rotted fruits. Data from natural infestation was used to compare with the behavior of CBB females from artificial infestation. However, data from natural infestations

**Table 1. Environmental variables at the UPR Agricultural Experimental Station in Adjuntas, Puerto Rico.**

| Temperature (°C) | | | | Humidity (%) | | | Rain (mm) | | |
|---|---|---|---|---|---|---|---|---|---|
| **Min** | **Max** | **Mean** | **Daily range**[a] | **Min** | **Mean** | **Max** | **Min** | **Mean** | **Max** |
| 12.0 | 34.1 | 21.8 ± 3.02 | 14.1 | 42.5 | 89.5 ± 0.07 | 100 | 0 | 0.02± 0.001 | 7.60 |

[a] Daily temperature range was calculated as maximum—minimum daily temperatures.

Mean ± S.E. and daily temperature range. These environmental variables were recorded using HOBO Pendant ® dataloggers and HOBO RX300 weather station.

were not used to determine demographic parameters and sensitivity analysis, since the time of infestation was unknown.

Temperature and light intensity were recorded continuously using Onset Hobo data loggers (HOBO Pendant ® Temperature and light, 64K, Onset Computer®, Bourne, MA, USA). Three data loggers were located in the middle of the plot, hung on coffee plants 1 m above the ground. Daily precipitation and humidity were obtained from a HOBO RX 3000 weather station (Onset Computer®, Bourne, MA, USA), located in a plot nearby. Data for environmental variables are shown in Table 1.

## Data analysis

The life cycle was divided into five developmental stages: egg, larva, pupa, juvenile and adult. Probability of surviving and remaining at the same life stage ($P_i$), probability of transition between stages, in which an individual survives and moves on to the next life stage ($G_i$), and fecundity of adult females ($F_i$) were calculated at the end of each evaluation, 50 days for lab and 56 days for field evaluations. Evaluation time differed between lab and field, because in the field the first juveniles and adults were observed at 56 days after infestation. While in the lab the first juveniles appeared at 28 days of evaluation; time was extended to 50 days to allow slower individuals to catch up, but no relevant changes were observed from day 45.

Fecundity ($F_i$) was calculated as the proportion of females that oviposited multiplied by the mean number of eggs produced by each female. $P_i$ probabilities were calculated only for CBB reared in the lab; for field evaluations these probabilities could not be determined since the same individuals were not followed through time.

Sampling protocol and frequency of field evaluations did not allow determination of the exact duration of each developmental stage. Data on developmental times from laboratory evaluations (Table 2) were used to calculate Gi or transitions between stages. Time of growth from egg to larva was estimated at 7 days; from larva to pupa 17 days; pupa to juvenile 7 days and juvenile to adult 4 days. To calculate $G_1$ (transition from egg to larva), the number of larvae counted at day 21 were divided by the number of eggs counted at day 14. Similarly, $G_2$ (transition from larva to pupa) was the proportion of pupae at day 42 divided by larvae in

**Table 2. Development time in days for different stages of the coffee berry borer (CBB, *Hypothenemus hampei*) reared on artificial diet Cenibroca.**

| Stage | Mean ± se | Minimum | Maximum | Confidence interval (95%) |
|---|---|---|---|---|
| Egg | 6.0 ± 0.2 | 2.0 | 10.0 | 5.6–6.5 |
| Larva | 17.2 ± 0.7 | 8.0 | 20.0 | 13.7–16.7 |
| Pupa | 6.3 ± 0.1 | 3.0 | 9.0 | 6.1–6.7 |
| Juvenile | 4.0 ± 0.1 | 6.0 | 3.0 | 3.8–4.3 |
| Egg to adult | 34.4 ± 0.7 | 21.0 | 40.0 | 32.9–35.9 |

Temperature 25°C (±1°C), 80–96% relative humidity (RH).

instar I counted at day 28. $G_3$ (transition from pupa to juvenile) was the proportion of juveniles counted at day 56 divided by the number of pupae at day 49. Finally, $G_4$ (transition from juveniles to adults) was determined as the proportion of adults at day 56 divided by the number of juveniles at day 56; the first juveniles were observed at day 56.

Staged-classified matrices were constructed for both lab and field evaluations [30]. The equation that projects the population forward in time is equals:

$$N_{t+1} = AN_t$$

where $N_t$ is a column vector representing the number of individuals in each developmental stage at time *t*: beginning of the evaluation. $N_{t+1}$ is the number of individuals at the different stages at time *t+1*: end of the evaluation period, 50 days for lab evaluations and 56 days for field. **A** represents the *n* x *n* population projection matrix, given as follows:

$$\begin{bmatrix} P_1 & 0 & 0 & 0 & F_1 \\ G_1 & P_2 & 0 & 0 & 0 \\ 0 & G_2 & P_3 & 0 & 0 \\ 0 & 0 & G_3 & P_4 & 0 \\ 0 & 0 & 0 & G_4 & P_5 \end{bmatrix}$$

Entries in the first row of the matrix represent egg survival ($P_1$) and fecundity ($F_1$) rates. In the case of the CBB only the adult female can oviposit and contribute to population growth ($F_1$). The main diagonal entries include the probabilities of surviving and staying in the same stage ($P_i$). The sub diagonal entries include the probability of surviving and advancing to the next stage ($G_i$). The zeros are transitions that do not occur in the life cycle of the CBB.

Matrices were used to calculate the following demographic parameters: population growth rate ($\lambda$), intrinsic rate of natural increase ($r_m$), mean generation time (T), doubling time (TD), and sensitivity values. Sensitivity analysis estimates the effect of changes in vital rates (e.g., survival, growth, fecundity) on population growth rate ($\lambda$). Sensitivity matrix contains the sensitivities of $\lambda_{max}$ to each element of the projection matrix $a_{i,j}$ [31], and is given by the equation:

$$S_{ij} = \frac{v_i w_i}{< w, v >} = \frac{\partial \lambda}{\partial a_{ij}}$$

Where **w** and **v** are the right and left eigen vectors of the projection matrix **A** (the stable stage distribution and reproductive value, respectively) and ‹**w,v**› denotes the scalar product of **w** and **v** [32].

## Perturbation analysis

Retroprospective analyses (LTRE or life table response experiment) were performed to determine which transitions or vital rates contributed to differences in values of $\lambda$ between populations of the CBB raised in the lab vs. CBB from artificial infestation in the field.

Prospective analyses were done to estimate the change necessary in vital rates to decrease the value of $\lambda$ to <1.0. To conduct prospective analysis, data from artificial infestation in the field were used for two reasons. First, the three vital rates with high sensitivities identified for both type of evaluations (field and lab) were the same (Table 4). Second, probabilities estimated for field evaluations reflect the true effects of biotic and abiotic variables on CBB populations.

Two types of analyses were conducted. First, all vital rates were modified including those with high sensitivity; in this case one vital rate was perturbed at a time holding the others

constant. Second, two or three vital rates with higher sensitivities were lowered simultaneously to find the point at which the population growth rate fell below one.

## Statistical analysis

The package popbio (version 2.4.3) was used to calculate all population parameters, generate elasticity matrices and perform perturbation analysis [33]. Generalized linear models (GLM) were used to compare female survival, presence, positions inside the fruit and population per fruit between CBB from artificial infestation vs. CBB from natural infestation in the field. Models for proportion of alive, dead, missing and positions of colonizing females were fitted using a quasibinomial error distribution. A Poisson error distribution was used for CBB total population per fruit models. GLM models with a quasibinomial distribution of errors were used to compare proportion of internal fruit rot according to survival, abandon rate and positions of colonizing females; data from fruit rot was transformed to proportion. Function (predict) was used to calculate the posterior model probabilities. All analyses were done in R 3.5.3. [34].

## Results

### Comparisons between artificial infestations vs. natural infestations in the field

A total of 468 coffee fruits were successfully infested by CBB females (*H. hampei*) raised under lab conditions. After infestation, these females behaved similarly to those in natural infestations. The proportion of missing, alive, and dead females for both types of infestations is shown in Fig 1. Neither the proportion of missing ($\chi^2_1 = 1.31$, $p = 0.149$) nor dead CBB females ($\chi^2_1 = 2.76$, $p = 0.069$) differed significantly between natural and artificial infestations. The only difference was in the proportion of live females, which was slightly higher for those from artificial infestation ($\chi^2_1 = 3.43$, $p = 0.009$); the most notable difference was observed in August, when the proportion of live females from natural infestation was 0.13 compared with 0.73 from artificial infestation (Fig 1A). In August, a higher proportion of missing females from natural (0.83) vs. artificial (0.21) infestations was also observed (Fig 1C).

Mortality from the fungus *B. bassiana* was not common. In only three fruits (<0.5%) the fungus was observed growing on a CBB adult female cadaver. No significant difference was observed in internal rot between natural vs. artificial infestation ($\chi^2_1 = 0.51$, $p = 0.340$). Percentages of internal rots were significantly higher in fruits in which the colonizing CBB was dead or missing than in fruits where the CBB was alive ($\chi^2_2 = 21.9$, $p < 0.0001$); differences in percentages of fruit rot between dead vs. missing CBBs were not significant ($p = 0.126$). Estimated fruit rot was 3.7% for fruits with live CBBs, 7.9% for dead CBBs and 12.5% for fruits with missing CBBs.

Penetration of the fruit by colonizing females progressed over the time for both types of infestation (Fig 2). Significant differences were found in positions C ($\chi^2_1 = 14.7$, $p < 0.0001$) and D ($\chi^2_1 = 20.3$, $p < 0.0001$) between artificial vs. natural infestations. Proportions of females in position C were higher for natural infestations (Fig 2C). Conversely, proportions of females in position D were higher for artificial infestations (Fig 2D). This means that more females reached the endosperm in naturally infested coffee, but they remained longer in this position (C) before starting to reproduce.

Coffee berry borer females from artificial infestation produced significantly more offspring per colonizing female than CBB females from natural infestation (Fig 3). The estimated abundance of offspring in all developmental stages was 20.6 per fruit for artificial infestation vs. 4.5 for natural ($\chi^2_1 = 5473$, $p < 0.0001$).

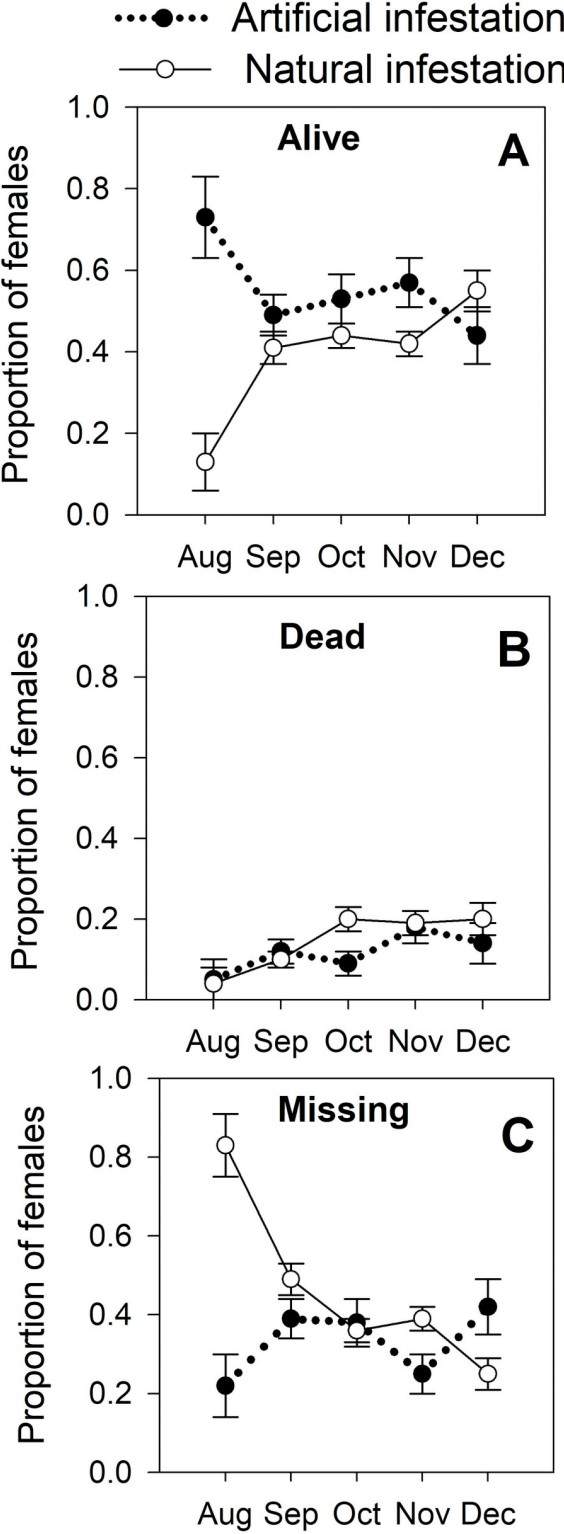

**Fig 1. Behavior of the coffee berry borer (CBB, *Hypothenemus hampei*) in artificial vs natural infestations in the field after penetration of coffee fruits: (A) proportion of alive females; (B) proportion of dead females; and (C) proportion of missing females.** Artificial infestation: coffee fruits in field were infested by CBB females reared in lab on artificial diet Cenibroca. Natural infestation: coffee fruits infested by CBB females in natural conditions. Sampling was done from August to December 2018 on a shade coffee plot of *Coffea arabica* cv. Limaní.

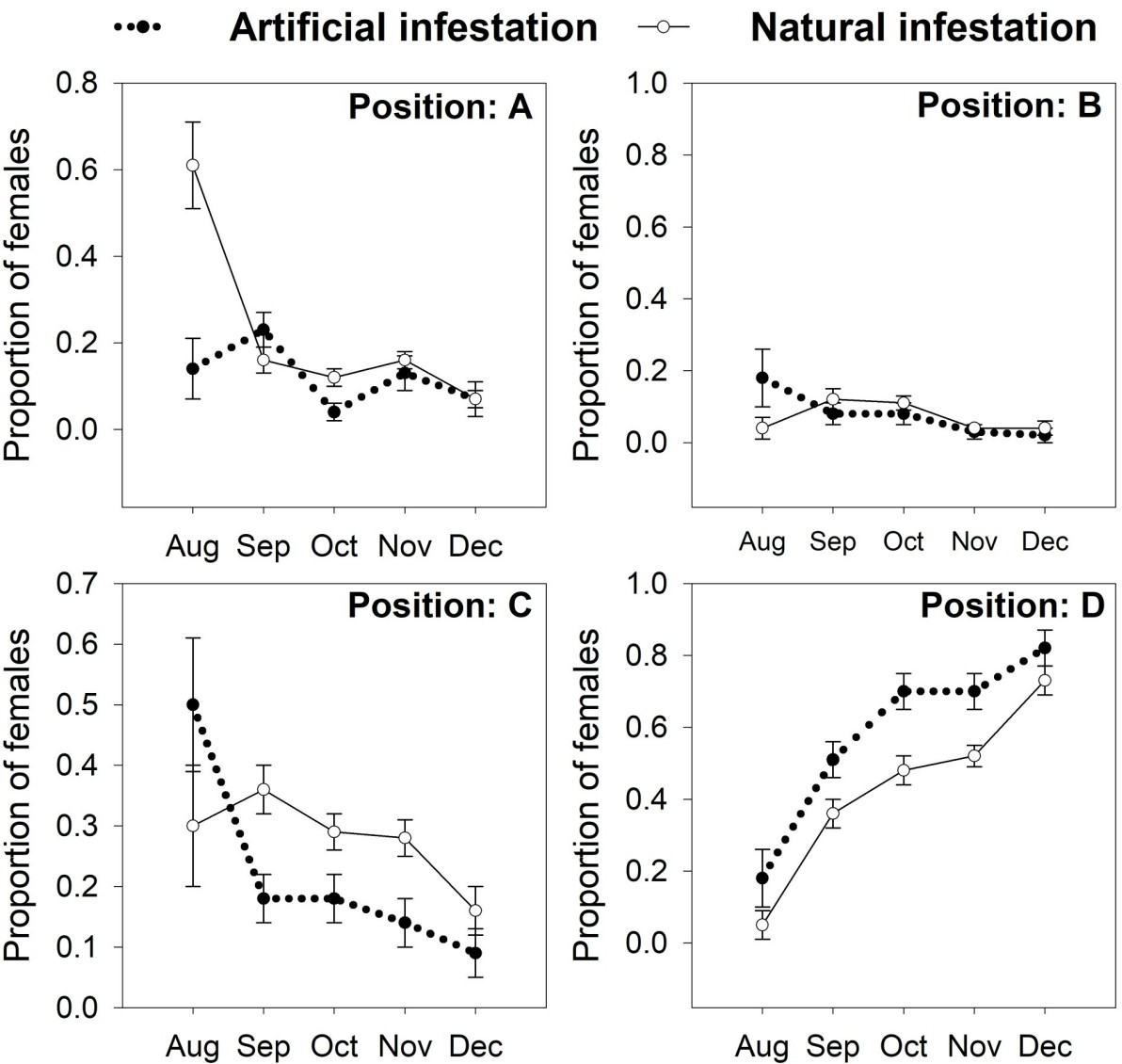

**Fig 2. Progress of the coffee berry borer (CBB, *Hypothenemus hampei*) in penetrating coffee fruits during the season.** Artificial infestations: coffee fruits in field were infested by CBB females raised in lab on artificial diet Cenibroca. Natural infestation: coffee fruits infested by CBB females in natural conditions. Sampling was done from August to December 2018 on a shade coffee plot of *Coffea arabica* cv. Limaní.

Estimated percentages of internal rots were significantly higher in fruits in which the colonizing female was in position C ($\chi^2_3 = 62.15$, $p < 0.0001$). Estimated values were 7.2% for females in position A, 1.0% for position B, 20.1% for position C and 2.9% for position D.

The abundance of individuals in each developmental stage for artificial infestation is shown in Fig 4. Eggs were found starting at day 7, and there were two peaks for eggs production at 14 and 35 days after infestation. Larvae had a similar pattern to eggs; they were always present and peaked at day 49. Pupae appeared at 42 days after infestation and peaked at day 49. Juvenile females and males appeared at same time, day 56. The first adult female offspring were observed on day 56 (the last sampling day). No adult males were observed. Developmental time from egg to adult under field conditions for a shade coffee plot with a mean temperature of 21.8 ˚C was approximately 49 days (Fig 4).

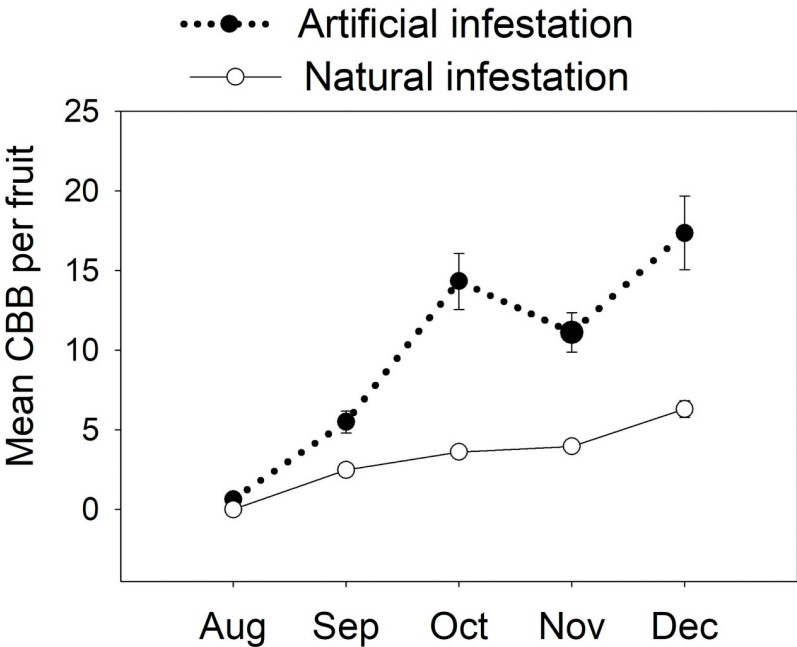

**Fig 3. Monthly changes in total number of individuals per fruit of the coffee berry borer (CBB,** *Hypothenemus hampei***) in artificial vs. natural infestations.** Means ± S.E. are shown. Artificial infestations: coffee fruits in the field were infested by CBB females raised in the lab on artificial diet Cenibroca. Natural infestation: coffee fruits infested by CBB females in natural conditions. Sampling was done from August to December 2018 on a shade coffee plot of *Coffea arabica* cv. Limaní.

**Life cycle developmental times and life table parameters.** Developmental time for CBB reared in the lab on the artificial diet Cenibroca at 25 ˚C ranged from 32.9 to 35.9 days. Mean developmental times per stage were: 6 days from eggs to larvae, 17.2 days from larvae to pupae including both instars (larva I and larva II), 6 days from pupae to juveniles and 4 days from juveniles to adults (Table 2).

The estimated population growth rate (λ) was over 1.0 for both populations (Table 3). Mean generation time (T) and intrinsic rate of natural increase ($r_m$) were higher for populations reared in the lab on the artificial diet. The time of duplication (DT) was almost halved for populations raised in the lab. This reflects the lab conditions: an artificial diet with high nutritional content, constant temperature and relative humidity, unlike conditions in the field (Table 3).

## Perturbation analysis

**Retroprospective analysis.** According to LTRE analyses, the reduction in values for $G_2$ (transition from larva to pupa) and $G_3$ (transition from pupa to juvenile) for CBB in the field made the largest contribution to the reduction in λ observed for these populations (Fig 5 and Table 3). Estimated transition or vital rates probabilities for populations raised in the lab and those in the field can be found in S1 and S2 Tables.

**Sensitivity analysis.** Probabilities of transitions between stages ($G_i$) were higher than probabilities to survive and remain at the same stage ($P_i$). The three transitions with largest potential impact on population growth rate (λ) were: survival rates of adults ($P_5$: 0.324 for lab and 0.389 for field populations), transition from larva to pupa ($G_2$: 0.316 for lab and 0.352 for field) and transition from pupa to juvenile ($G_3$: 0.345 for lab and 0.515 for field) (Table 4).

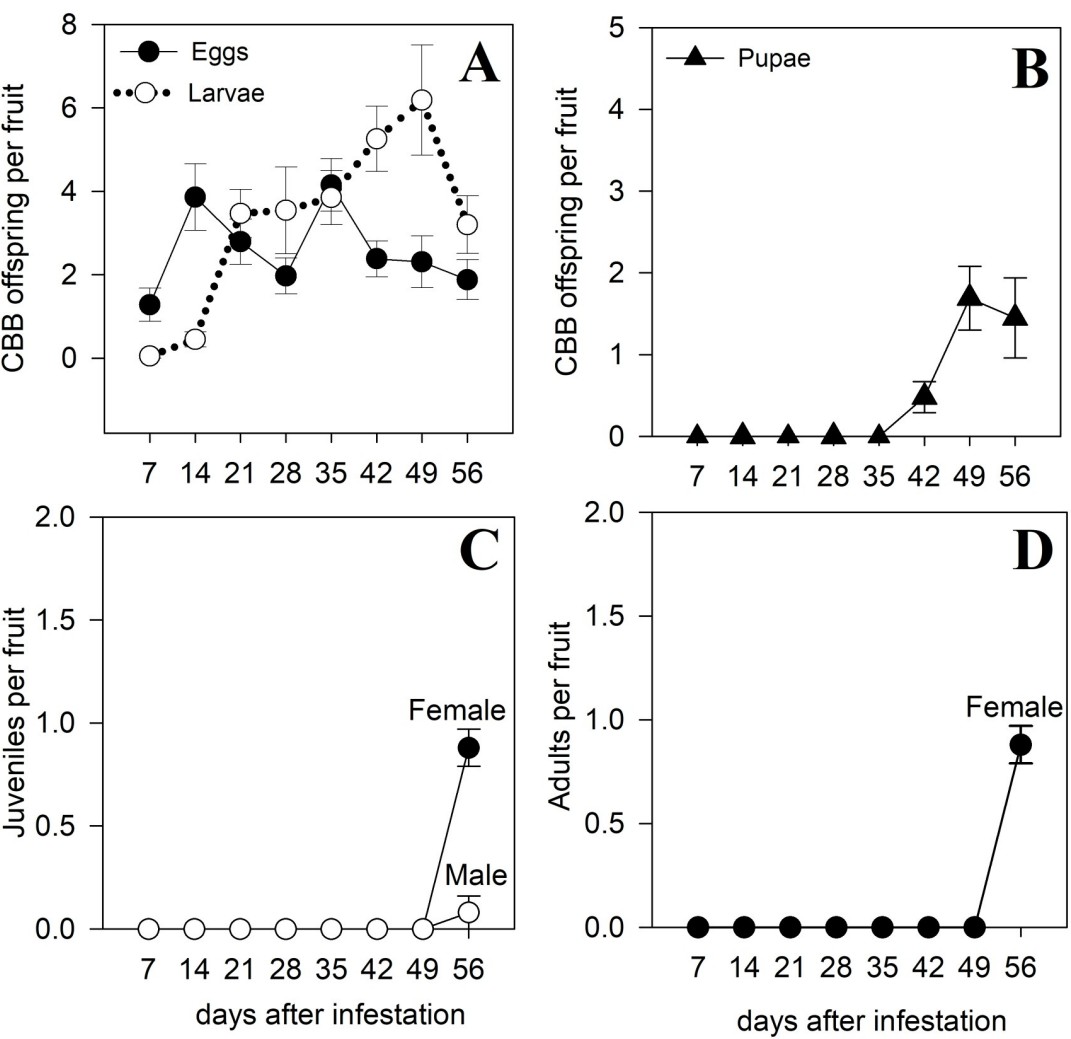

**Fig 4. Number of coffee berry borer (CBB, *Hypothenemus hampei*) per fruit in different developmental stages following artificial infestation:(A) eggs and larvae; (B) pupae; (C) juveniles; and (D) adults.** Coffee fruits were infested by CBB females raised in the lab on artificial diet Cenibroca. Means ± S.E. are shown. Sampling was done from August to December 2018 on a shade coffee plot of *Coffea arabica* cv. Limaní.

**Table 3. Life table parameters for populations of the coffee berry borer (CBB, *Hypothenemus hampei*) reared in the lab on artificial diet (Cenibroca) vs. artificial infestations in the field made on *Coffee arabica* cv. Limaní.**

| Cohort | Life Table Parameters | | | |
|---|---|---|---|---|
| | $\lambda$ | T | $r_m$ | DT |
| Lab [a] | 1.74 | 41.5 | 0.058 | 11.8 |
| Artificial infestations [b] | 1.31 | 40.1 | 0.053 | 13.1 |

$\lambda$ = population growth rate; T = mean generation time; $r_m$ = intrinsic rate of natural increase: DT = doubling time (days)

[a] = individuals reared in the lab on the artificial diet Cenibroca (Temperature: 25 ± 1˚C, RH 80–96%).

[b] = Individuals in the field on *Coffea arabica* cv. Limaní following artificial infestations (Mean temperature 21.8 ± 3˚C, RH 42.5–100%).

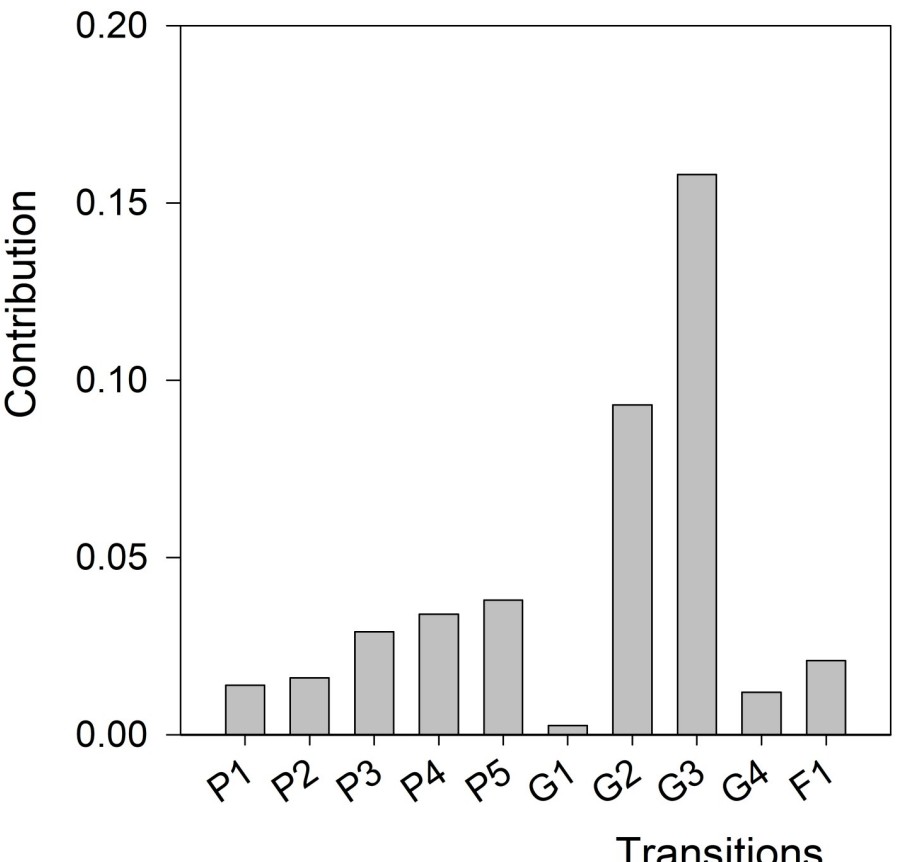

**Fig 5. Life table response experiment (LTRE) analysis of the coffee berry borer (CBB, *Hypothenemus hampei*) reared in the lab vs. CBB in the field (artificial infestations).** The bars show the contribution of each life transition to the reduction in population growth rate (λ) when comparing between populations. $G_i$: probability of transition between stages, $P_i$: probability to survive and remain in the same stage, $F_i$: fertility of adult females. (1) Eggs, (2) Larvae, (3) Pupae, (4) Juveniles and (5) Adults.

**Prospective perturbation analyses.** The effects of proportional changes in vital transitions on the population growth rate (λ) for CBB populations in the field are shown in Figs 6 and 7 and Table 5. When only one parameter is perturbed at a time, including vital rates with the highest sensitivities ($G_2$, $G_3$ and $P_5$), no important changes in population growth rate were observed. When ($G_i$) (transition from egg to larva) was reduced by ≥70%, λ was < 1.0 (0.98), but for the remaining transitions the population growth rate was > 1.0 even when probabilities were reduced by ≥ 90% (Fig 7).

Significant effects on population growth rate (λ) are possible when varying two or three transitions with higher sensitivities: that is, when $G_2$, $G_3$ and $P_5$ are perturbed at the same time (Fig 7 and Table 5). When two transitions are perturbed at the same time reduction of 30%–35% in both parameters is necessary to reduce the value of λ to < 1.0. For instance, a change of -35% for $G_2$ (0.60 to 0.25) and $G_3$ (0.40 to 0.05) produced a decrease of 30.3% in λ (1.32 to 0.92) (Fig 7A). Perturbing $P_5$ and $G_3$ at the same time, a 30% of reduction for each vital rate is necessary to decrease λ from 1.32 to 0.96; $G_3$ was the transition with highest sensitivity (Fig 7C).

When the three transitions are perturbed at the same time, the disturbance necessary to produce the same decrease in λ (1.32 to 0.96) is reduced to 25%. $P_5$ (adult survival) should be

**Table 4. Sensitivities and elasticities for the coffee berry borer (CBB, *Hypothenemus hampei*) reared in the lab on the artificial diet (Cenibroca) and in the field, artificial infestations made on *Coffeea arabica* cv. Limaní.**

| Vital rate | Lab | | Field | |
|---|---|---|---|---|
| | Sensitivities | Elasticities | Sensitivities | Elasticities |
| $G_1$ | 0.299 | 0.154 | 0.225 | 0.049 |
| $G_2$ | **0.316** | 0.154 | **0.352** | 0.049 |
| $G_3$ | **0.345** | 0.154 | **0.515** | 0.049 |
| $G_4$ | 0.240 | 0.154 | 0.267 | 0.049 |
| $P_1$ | 0.163 | 0.008 | - | - |
| $P_2$ | 0.164 | 0.009 | - | - |
| $P_3$ | 0.172 | 0.017 | - | - |
| $P_4$ | 0.175 | 0.021 | - | - |
| $P_5$ | **0.324** | 0.169 | **0.389** | 0.703 |
| $F_1$ | 0.023 | 0.154 | 0.019 | 0.049 |

$G_1$: probability of transition between stages, $P_i$: probability to survive and remain in the same stage, $F_i$: fertility of adult females. (1) Eggs, (2) Larvae, (3) Pupae, (4) Juveniles, (5) Adults. The sampling protocol for field evaluations did not allow us to determine the probability to survive and remain at the same stage ($P_i$) for some developmental stages (eggs, larvae, pupae and juveniles). Transitions with largest potential impact on population growth rate ($\lambda$) are shown in **bold**.

lowered from 0.80 to 0.55; $G_2$ (transition from larva to pupa) from 0.60 to 0.35, and $G_3$ (transition from pupa to juvenile) from 0.40 to 0.15 (Table 5).

## Discussion

The principal aims of this study were to collect demographic data to perform retrospective and prospective perturbation analyses of coffee berry borer (CBB) populations. Prospective analyses are based on the determination of vital rates with highest sensitivities, which can be perturbed in order to decrease the value of population growth rate ($\lambda$) to < 1.0. Identification of vital rates with highest contributions to $\lambda$ can be powerful tools to improve Integrated Pest Management (IPM) programs [18, 35].

Field and laboratory evaluations were conducted to determine demographic parameters for CBB populations, the reproductive potential of the CBB on the Cenibroca artificial diet is well known [26, 36–38]; however, few studies have conducted field experiments to obtain demographic data [21, 24]. To the best of our knowledge, this is the first study to conduct field evaluations of the CBB to perform prospective perturbation analysis. Three vital rates were identified as important targets for CBB management: $G_2$ (transition from larva to pupa), $G_3$ (transition from pupa to juvenile) and $P_5$ (survival of adult females); these vital rates had the highest sensitivities, and their perturbation is biologically feasible (Table 4).

Perturbation analysis indicated that an efficient management for the CBB should not be focused on only one developmental stage. When one vital rate was perturbed at a time and the others remained constant, high values of change in each vital rate were necessary to reduce the value of $\lambda$ to < 1.0. $G_2$ and $G_3$ had to be reduced more than 90% (Fig 6A); for $P_5$, not even a reduction of 100% reduced the value of $\lambda$ to < 1.0 (Fig 6B). However, a significant reduction of $\lambda$ to < 1.0 is possible when two or three of these vital rates ($G_2$, $G_3$ and $P_5$) are perturbed at a same time (Table 5).

Several methods, including biological, cultural and chemical control, have been tested and recommended to reduce CBB populations [3, 7, 39]. Biological control organisms include parasitoid wasps [40–42], entomopathogenic fungi (*B. bassiana* and *Metharizium anisopliae* (Metchnikoff) Sorokin) [43–46], nematodes [47, 48], and ants [49–51].

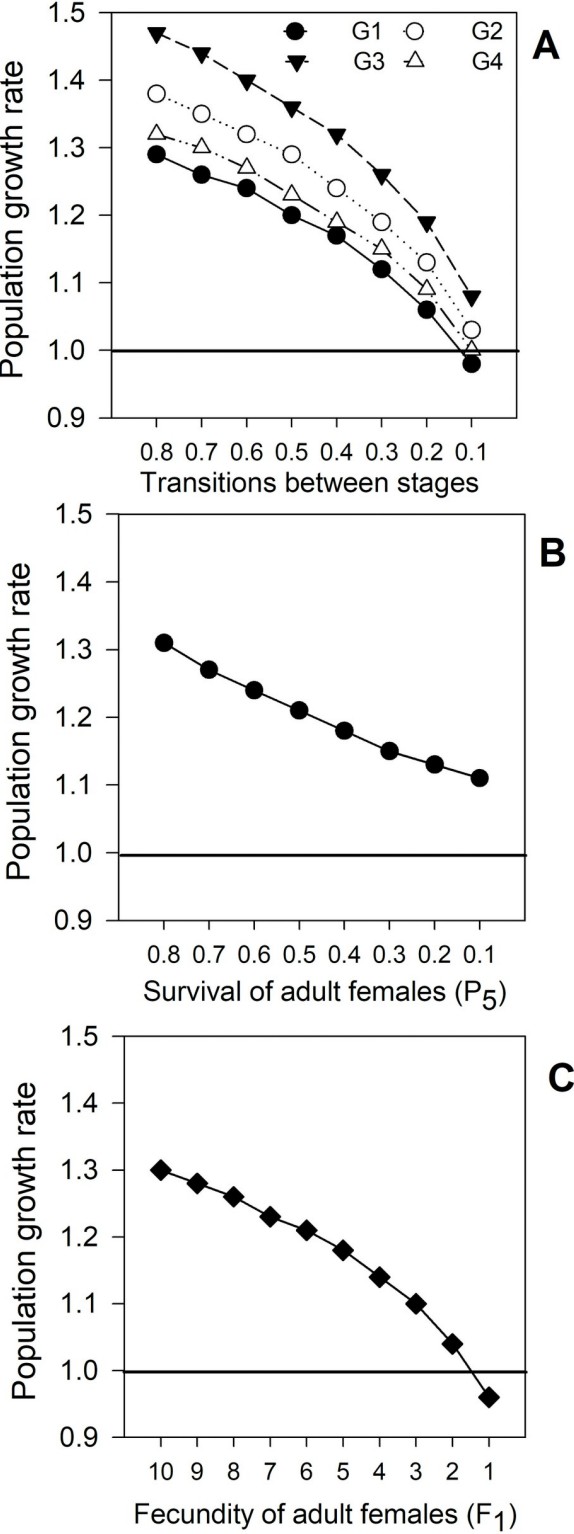

**Fig 6. Coffee berry borer (CBB, *Hypothenemus hampei*) population growth rate (λ) as a function of: (A) Probability of transition between stages, ($G_1$) eggs, ($G_2$) larvae, ($G_3$) pupae and ($G_4$) juveniles; (B) survival of adult females ($P_5$) and (C) fecundity of adult females ($F_1$), perturbing only one vital rate at a time.** Data from field evaluations. λ value before the perturbations was 1.32.

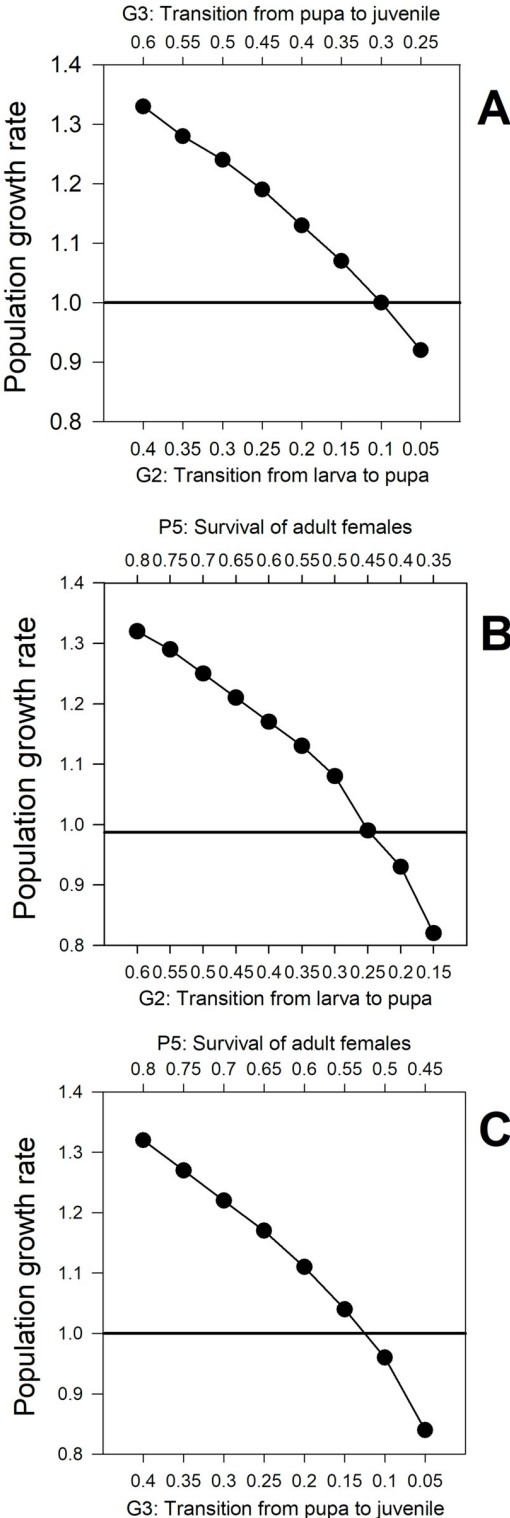

**Fig 7. Coffee berry borer (CBB, *Hypothenemus hampei*) population growth rate (λ) perturbing two transitions with higher sensitivities at the same time: (A) Perturbing $G_2$ and $G_3$; (B) Perturbing $G_2$ and $P_5$; (C) Perturbing $G_3$ and $P_5$.** Data from field evaluations of artificial infestations on shade coffee plot *C. arabica* cv. Limaní. λ value before the perturbations was 1.32.

**Table 5. Coffee berry borer (CBB, _Hypothenemus hampei_) population growth rate (λ) perturbing three transitions with higher sensitivities at the same time.**

| $G_2$ | $G_3$ | $P_5$ | λ |
|------|------|------|------|
| 0.60 | 0.40 | 0.80 | 1.33 |
| 0.55 | 0.35 | 0.75 | 1.27 |
| 0.50 | 0.30 | 0.70 | 1.20 |
| 0.45 | 0.25 | 0.65 | 1.13 |
| 0.40 | 0.20 | 0.60 | 1.05 |
| **0.35** | **0.15** | **0.55** | **0.96** |
| 0.30 | 0.10 | 0.50 | 0.86 |
| 0.25 | 0.05 | 0.45 | 0.74 |

$G_2$: Transition from larva to pupa; $G_3$: Transition from pupa to juvenile, and $P_5$: Survival of adult females. Data from field evaluations of artificial infestations on _C. arabica_ cv. Limaní. λ value before the perturbations was 1.32.

Ants and parasitoid wasps could help to reduce probabilities $G_2$, $G_3$ and $P_5$ which deal with larvae, pupae and adult females, respectively. Some lab studies have demonstrated that several species of ants have great capacity to remove immature stages of the CBB from different containers [52, 53]. But in the field their predatory capacity is significantly reduced, due to the difficulty of entering coffee fruits through the hole made by the CBB; some ants are too large to enter the infested fruits [53]. Only small species like _Pheidole radoszkowskii_ (Mayr), _Crematogaster crinosa_ (Mayr), _Wasmannia auropunctata_ (Roger) and _Solenopsis picea_ (Emery) have the capacity to penetrate and remove immature stages of the CBB from coffee fruits.

_W. auropunctata_, _S. picea_ and other species of ants like _Solenopsis invicta_ (Buren) and _Tetramollium simillum_ (Roger) also have great capacity to predate adult females and prevent their colonization of coffee fruits [50, 51, 54, 55]. Another important predator of the CBB is the flat bark beetle _Cathartus quadricollis_ (Guérin-Méneville) [56]; laboratory assays demonstrated the capacity of adults and larvae of this species to feed on all stages of CBB [56, 57].

Four species of parasitoid wasps have been identified as natural enemies of the CBB, _Cephalonomia stephanoderis_ (Betrem), _Prorops nasuta_ (Waterson), _Phymastichus coffea_ (La Salle) and _Heterospilus coffeicola_ (Schmiedknechtcan parasitize CBB adult females. Additionally, _C. stephanoderis_, _P. nasuta_ and _H. coffeicola_ can predate immature stages of the CBB, including big larvae and pupae [40, 41, 58].

Fungi also are important natural enemies of the CBB and can affect the survival of adult females. One of the most common fungi used to manage the CBB is _B. bassiana_. In Puerto Rico and other countries _B. bassiana_ is commonly applied as Mycotrol® or Botanigard®. However, natural infections with local strains of _B. bassiana_ have been observed. Reported rates of infection range from <1 to 70% of CBB females that were infesting the coffee fruits [44, 59–62]; Gallardo et al. [60] and Vega et al. [59] observed < 1% of natural infections as reported here. Both commercial products and local isolates of _B. bassiana_ are efficient reducing the damage caused by the CBB.

Like ants, the efficiency of _B. bassiana_ is higher under laboratory conditions. Reports of mortality of CBB females ranged from 52 to 100%, and 25 to 98% [63, 64]. In the field this efficiency is affected by environmental conditions, mainly temperature, relative humidity and solar radiation. For example, Wraight et al. [65] reported percentages of CBB mortality from local _B. bassiana_ isolates from 0.7 to 28%, and similar mortality was reported with the strain GHA from the commercial products Mycotrol® and Botanigard® [12]. However, in the field local strains survived and persisted better than the GHA strain derived from Mycotrol® [64].

Other pathogenic fungi that cause internal rots could be an additional factor for CBB mortality, or stimulate its migration to other fruits [6, 21]. Likewise, we observed more internal rot in fruits in which CBB females were found dead or missing compared with those fruits in which females were alive. It is likely that some of the rot pathogens killed the CBB or induced them to abandon the fruits. Four species of *Colletotrichum* (*C. fructicola*, *C. siamense*, *C. theobromicola* and *C. tropicale*) were recently reported as pathogens of coffee fruits, with the ability to cause internal and external rots; *Fusarium* was also isolated from internal rots [10]. But it is unknown if these fungi that cause coffee fruit rot are also pathogens for the CBB.

Survival of CBB adult females can also be reduced using cultural methods, mainly sanitation, that is, the periodic collection of ripe and raisin fruits during and after the harvest [39]. Fruits remaining on the plants and on the ground are important reservoirs for the CBB [3, 6, 66, 67]. In places where there is a single coffee crop per year, like Puerto Rico and Hawaii, sanitation should be mainly focused on fruits remaining on the plants, which can harbor higher populations of the CBB than fruits on the ground [67, 68]. For example, in Puerto Rico a single raisin fruit remaining on a tree can harbor up to 94 CBBs, of which 45 were adult females, whereas fruits remaining on the ground decomposed quickly [68]. Implementation of sanitation measures can contribute up to 80% of successful management the CBB [69].

Chemical control using insecticides is limited in many coffee producing countries due to health and environmental concerns. Additionally, the frequent use of an insecticide leads to development of resistance; for instance, resistance of the CBB to endosulfan has been reported, and endosulfan is one of the most used insecticides against the CBB [70].

Higher values for the demographic parameters intrinsic rate of natural increase ($r_m$) and population growth rate ($\lambda$) were observed in populations reared in the lab than in populations in the field from artificial infestations. Overall, populations growing in the lab with constant abiotic variables (temperature and RH), few disturbances and an artificial diet with high nutritional content performed better (Table 3) than field populations. Retrospective (LTREs) analyses showed that reduction in $G_2$ (transition from larva to pupa) and $G_3$ (transition from pupa to juvenile) probabilities made the largest contribution to the reduction in $\lambda$ observed for populations in the field (Fig 5 and Table 3). Prospective analysis also identified these two transitions as important targets for CBB management; these vital rates had higher values of elasticity.

CBB females expulse immature stages from the fruit under some circumstances, and they also conduct a rigorous hygiene of dead immature stages, which are removed rapidly from the fruit [21]. This report agrees with our results: we observed a marked decrease in the number of pupae compared with number of larvae counted. For instance, 297 larvae in instar I resulted in 147 pupae, suggesting that 51% of larvae died, were predated by ants and/or were expulsed from the fruit by adult CBBs before they became pupae. A more drastic loss was observed for $G_3$: 177 pupae resulted in only 24 juveniles; this means that 84% of pupae disappeared from the fruits. However, these decreases in $G_2$ and $G_3$ probabilities under natural conditions were not sufficient to reduce the value of $\lambda$ to <1.0.

Several studies have determined demographic parameters of CBB populations under lab and field conditions. Most of the lab studies have used the artificial diet Cenibroca to grow the CBB. These studies have evaluated the effect of temperature ranging from 23 to 29 ˚C, modifications of diet composition and continuous rearing for several generations [26, 37, 38, 71].

These studies reported the following ranges for the four life table parameters that we determined in this study: $r_m$: 0.057–0.078, $\lambda$: 1.059–1.081, DT: 8.96–12.16 and T: 37.97–47.66. These parameters were affected by temperature and number of generations. In the case of temperature, the values of $r_m$ and $\lambda$ increased with temperature, while values of DT and T decreased with temperature; at higher temperatures the CBB performed better and required less time to

reproduce. These studies concluded that the optimum temperature for mass rearing of the CBB is 27 °C [26, 71]. However, in our case we raised the CBBs at 25 °C to avoid the diets in the tubes from drying out or losing humidity too fast; salt solutions were not used to control humidity inside the growth chamber as recommended by Portilla [72].

In the case of continuous rearing Portilla et al. [37] found that performance of the CBB declined with increasing number of generations; all life table parameters decreased. The effect of temperature and continous rearing, could be two possible explanations for the differences in the values of life table parameters reported in these studies vs. the parameters reported here. We reared the CBB on Cenibroca diet at 25 ± 1°C and 80–96% relative humidity with a CBB population from the 6[th] generation in culture, and our life table parameters were: $r_m$: 0.058, λ: 1.74, DT: 11.8 and T: 41.5 days. These values for $r_m$ and DT were similar to those reported by Portilla et al. [37] for their 5[th] generation ($r_m$: 0.057, DT: 12.16). However, our value for λ is higher than those of all these studies; this means that our CBB populations increased more during our period of evaluation (50 days), despite the $r_m$ or increased rate of natural increase is declining over generations.

Relatively few studies have determined demographic parameters of CBB populations in the field. Baker *et al.* [21] and Ruiz and Baker [24] conducted experiments on coffee farms in Mexico and Colombia, respectively. Values reported in Mexico, with a mean temperature of 26°C, were: λ = 1.067, $r_m$ = 0.065, T = 44.7 days and DT = 10.7 days. In Colombia with a mean temperature of 20.7–21.6°C, values estimated were: λ = 1.060 to 1.070, $r_m$ = 0.053–0.066, T = 48.5 to 54.3 days and DT = 10.5 to 10.7 days. Values of $r_m$, T and DT were similar to those estimated in this study in Puerto Rico with a mean temperature 21.8°C ($r_m$ = 0.053, T = 40.1 and DT = 13.1 days, Table 3). However, we observed a difference in the value of λ, which was higher for our populations (1.32); this value corresponds to artificial infestations in the field. To perform these infestations, we used adult female CBBs reared in the lab on the artificial diet, which had the best performance, as mentioned above. One possible explanation could be related to the fecundity of CBB females, which was significantly higher for females reared in the lab vs. those that infested naturally (Fig 3). Explanations for this difference in fecundity are discussed in more detail in the next section.

The first adult offspring in the field were observed 56 days after infestation, which suggests that developmental time from egg to adult was approximately 49 days. These results contrast with previous field observations made at similar mean temperatures. In a shade plot in Mexico with a mean temperature of 20.8°C and planted with *C. canephora* cv. Robusta, the first adults appeared between 42 to 49 days after infestation [21]. In Hawaii, in a plot planted with *C. arabica* cv. Typica exposed to full sun, with mean temperature of 20.6 °C (like that reported by Baker et al., [21]) the first adults appeared 47 days after infestation; while at 22.0°C (similar to that reported in this study), first adults appeared 43 days after infestation [73].

Hamilton *et al.* [73] suggested that fluctuations in daily temperatures can also influence developmental time: sites with higher daily ranges in temperatures had slower CBB development. For instance, in a site with a daily range of temperature of 14.9°C the first adults appeared at 51 days after infestation. Our results support these observations: the daily range in temperature was 14.1°C in this study, approximately 0.8 °C less than that reported by Hamilton *et al.* [73], and developmental time was longer (the first adults appeared 56 days after infestation). These results suggest that mean temperature is not the only factor that influences developmental time; there are other factors such as daily range in temperature.

Artificial infestations by CBB adult females reared in the lab were successful: 468 females infested and reproduced inside coffee fruits. After infesting the fruits, behavior of these females was like that of females that infested coffee naturally. Females reared in the lab and those from

the field were affected equally by abiotic (temperature and relative humidity) and biotic variables (ants, parasitoids, entomopathogenic fungi); no differences were observed in death and abandonment rate (Fig 1). The only difference was in the proportion of missing females in August at the beginning of the experiment, which was higher from natural (0.83) vs. artificial (0.21) infestations (Fig 1C). This low rate of abandon from artificial infestations may be a consequence of the short time that females were exposed to environmental or biological factors that kill them or stimulate their migration to other fruits; only a week had passed since the artificial infestation were done. While, in natural infestations some females abandoned the fruits before the experiment began.

Other difference in behavior was observed in the proportion of females in position C and D, more females from artificial infestation were in position D. In other words, females reared in the lab were more likely to reach the endosperm and start to reproduce there (Fig 2). This high proportion of colonizing females in position D could be related to the frequency of males. When CBB individuals were reared in the artificial diet Cenibroca male offspring were found in all vials evaluated, and the number of males per vial increased through generations from $2.4 \pm 0.3$ ($F_1$) to $11.3 \pm 0.3$ ($F_{10}$) [25]. Higher frequency of males in the artificial diet could increase the possibility that more females are fertilized, reach the endosperm, and oviposit. In the field, Mariño et al. [74] evaluated CBB reproduction and found that 61% of fruits did not contain males (this percentage only include fruits with juveniles and adults, which are the stages that can be sexed).

CBB females reared in the lab and used to perform artificial infestations were more fertile, producing almost 4x more offspring per fruit that CBB females from natural infestations (Fig 3). This high fecundity could be associated with the food source and quality, and the temperature during the pre-oviposition period to which these females were exposed.

Food source, availability and quality have a direct impact on the growth, development, and reproduction of insects. In the case of herbivorous insects like the CBB, successful development and reproduction depends on the nutrients offered by its host plant. One of the most important elements that determines host quality is nitrogen [75, 76]. Several studies have demonstrated that the amount of nitrogen available during the development of larvae, pupae and adults has important effects on the future fecundity and fertility [75]. For example, females of the tortoise beetle *Paropsis atomaria* (Oliver) that feed on foliage of *Eucalyptus blakelyi* (Maiden) with high levels of nitrogen (4.12%) produced 5x more eggs than females that feed of foliage with low levels of nitrogen (1.15%) [77].

The artificial diets used to grow insects are usually a mixture of nutrients (carbohydrates, lipids and proteins), vitamins, salts and preservatives [78]. The artificial diet Cenibroca that we used to grow CBB populations contains casein and yeasts [36]. Casein and yeasts are important sources of nitrogen. The N content of casein is 14.2% [79], almost 10x higher than N content of endosperm from coffee fruits, which is 1.89% [80].

Finally, these higher proportions of females in position D from females reared on the artificial diets could also be related to temperature during the pre-oviposition period to which these females were exposed. CBB adult females reared on artificial diet and used to perform artificial infestations were maintained at a constant temperature of 25˚C, while females from natural infestation were exposed to variables temperatures from 12˚C to 34.1˚C, with a daily range 14.1˚C (Table 1). Jaramillo et al. [23] evaluated the effect of temperature on fecundity of CBB females reared in the lab on fruits of *C. arabica* cv. Ruiri at eight temperatures ranging from 15˚C to 35˚C. Females exposed to cold temperatures (15˚C) were able to reach the endosperm but did not oviposit, while females exposed to warm temperatures (35˚C) did not reach the endosperm. Bergamin [81] also suggested that exposure to low temperatures had a negative effect on the oviposition activity of CBB females.

## Conclusions and recommendations

This study is the first to use artificial infestations of the CBB in the field to perform elasticity and prospective perturbation analyses. These analyses identified which vital rates or CBB developmental stages are the best targets to improve CBB management programs. Results from elasticity analyses determined that transition from larvae to pupae, transition from pupae to juvenile and survival of adult females are the three vital rates that most influenced population growth rate (λ). Perturbation analyses showed that a successful Integrated Pest Management (IPM) program for the CBB should not be focused only in one stage or vital rate. When one vital rate including the three with highest sensitivity were perturbed at a time and the others remained constant, very high percentages of perturbation were necessary to reduce the value of λ to <1.0.

IPM programs should be directed to enhance the diversity of natural enemies, mainly ants, parasitoids and predators, which have the capacity to affect or impact the survival of these three important targets. To increase the diversity of natural enemies of the CBB, the first steps could be the re-introduction of shade trees and the reduction or elimination of chemical pesticides used to manage the CBB and other coffee pests like the coffee leaf miner (*Leucoptera coffeela* Guérin-Méneville), which also reduce populations of natural enemies of the CBB.

Future investigation and experimentation are needed to determine the effects of natural enemies on the CBB populations. It is important to focus on the role of these organisms as predators of larvae, pupae and CBB adult females, and to determine the impact of natural enemies on the demographic parameters of the CBB, one possible experiment to determine this effect could be the exclusion of natural enemies using caged and uncaged coffee trees.

## Supporting information

**S1 Table. Projection matrix for populations of the coffee berry borer (CBB, *Hypothenemus hampei*) reared in lab on the artificial diet CENIBROCA at temperature of 25 ± 1˚C, relative humidity 80–96%.**
(PDF)

**S2 Table. Projection matrix for populations of the coffee berry borer (CBB, *Hypothenemus hampei*) artificial infestations in a shade coffee plot planted with *Coffeea arabica* cv. Limaní, mean temperature 21.8 ± 3˚C, relative humidity ranged from 42.5 to 100%.**
(PDF)

## Acknowledgments

We thank Rocío Rivera for her help in the lab.

## Author Contributions

**Conceptualization:** Yobana A. Mariño, Alberto M. Sabat.

**Formal analysis:** Yobana A. Mariño, Alberto M. Sabat.

**Funding acquisition:** Paul Bayman.

**Investigation:** Yobana A. Mariño.

**Methodology:** Yobana A. Mariño, Alberto M. Sabat.

**Project administration:** Paul Bayman.

**Writing – original draft:** Yobana A. Mariño.

**Writing – review & editing:** Paul Bayman, Alberto M. Sabat.

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
