## [Decision Letter · Decision Letter 0]

9 Aug 2021

PONE-D-21-18910

Demography and perturbation analyses of the coffee berry borer *Hypothenemus hampei* (Coleoptera: Curculionidae): implications for management

PLOS ONE

Dear Dr. Marino,

Thank you for submitting your manuscript to PLOS ONE. After careful consideration, we feel that it has merit but does not fully meet PLOS ONE’s publication criteria as it currently stands. Therefore, we invite you to submit a revised version of the manuscript that addresses the points raised during the review process.

Please carefully follow the rewievers' requests, that you can find in this decision lettere and in the attached files, in which specific comments and corrections were provided by the reviewers.

We look forward to receiving your revised manuscript.

Kind regards,

Patrizia Falabella

Academic Editor

PLOS ONE

Journal Requirements:

"We thank Rocío Rivera for help in the lab. This project was supported by USDA ARS Specific Cooperative Agreements 58-1245-4-082 and 58-1245-4-083 and by an Advanced Research Grant from the Puerto Rico Science and Technology Research Trust."

"PB, USDA ARS Specific Cooperative Agreements 58-1245-4-082 and 58-1245-4-083.

Reviewers' comments:

Reviewer's Responses to Questions

**Comments to the Author**

1. Is the manuscript technically sound, and do the data support the conclusions?

Reviewer #1: Yes

Reviewer #2: Yes

2. Has the statistical analysis been performed appropriately and rigorously? 

Reviewer #1: Yes

Reviewer #2: Yes

3. Have the authors made all data underlying the findings in their manuscript fully available?

Reviewer #1: Yes

Reviewer #2: Yes

4. Is the manuscript presented in an intelligible fashion and written in standard English?

Reviewer #1: Yes

Reviewer #2: Yes

5. Review Comments to the Author

Reviewer #1: This paper reported a comprehensive study on demographic parameters values on CBB rearing on artificial diet under laboratory conditions and artificial infestation under field conditions. The study was well designed and conducted. The results were properly analyzed. However, the presentation of the results and the discussion needs to be improved.

This study demonstrated that biological control agents and physical conditions could affect the CBB's demographic values. This information is crucial for the CBB control. Also is demonstrating that the rm values of the CBB under field conditions could be much higher that the ones that have been reported in previous studies.

Reviewer #2: The study by Marino et al. describes CBB development times under natural and laboratory conditions and conducts perturbation and elasticity analyses to determine which stages in the life cycle contribute most to population growth rate. The study is well-thought out, nicely written, the analyses seem appropriate, and the discussion is comprehensive. Overall, this represents an important contribution to the literature on this worldwide pest of coffee. I have some minor comments which should easily be addressed. I enjoyed reading this manuscript and look forward to seeing it published.

L87-88: How did you decide on the environmental conditions to maintain the lab colony (why 25 C and 80-96% RH)?

L124-125: Were entomological sleeves placed around the entire branch? If not, how do you know that artificial infestations were the result of lab-reared CBB and not naturally occurring CBB from neighboring trees/branches?

L128: You might define larvae I vs. larvae II and pre-pupae vs. pupae.

L135-138: Please explain why mortality by Bb and internal fruit rot was evaluated and how these variables relate to the overall goals of the study.

L141: “Fully feral” is the same as natural, correct? I think best to stick to “natural field-infestation” so as not to confuse the reader (it could be inferred that these were infestations in wild coffee).

L1579-159: Are these daily temp and RH ranges, or ranges across the entire sampling period?

Table 1: Why is SE only shown for mean temp?

L172: Why were lab populations followed for 50 days and artificially infested populations followed for 56 days? Was this to account for the two days over which infestation could have occurred for artificial infestations?

L189: Here you say lab populations were evaluated at 56 days not 50 days?

L270-275: This should be moved to discussion.

L327: How could both juveniles and adults show up on same day (day 56)? Wouldn’t the presence of adults on day 56 mean that juveniles had to be present before that? Does this mean you might have missed the first appearance of juveniles?

Table 2: Change “media” to “mean”. Please include SE for min and max.

L523, L528: Change “parasite” to “parasitize”.

L526: Add “other” before predators.

L559-560: Yes, but this is because B. bassiana was not sprayed during your study, correct? Please include commonly reported rates of mortality due to B. bassiana from other studies.

L587: change to “more often recommended”.

L588: This is also true for other regions, such as Hawaii (see Johnson & Fortna et al. 2019).

L602: add “adult”

L648: Please explain why coffee species and variety would affect CBB development times.

L691: I am confused by the statement “endosperm or almond from coffee fruits”. Please revise.

L704: Exposition? Do you mean exposure?

6. PLOS authors have the option to publish the peer review history of their article (what does this mean?). If published, this will include your full peer review and any attached files.

Reviewer #1: No

Reviewer #2: No

---

## [Author Response · Author response to Decision Letter 0]

27 Sep 2021

Reviewer #1. 

The study by Marino et al. describes CBB development times under natural and laboratory conditions and conducts perturbation and elasticity analyses to determine which stages in the life cycle contribute most to population growth rate. The study is well-thought out, nicely written, the analyses seem appropriate, and the discussion is comprehensive. Overall, this represents an important contribution to the literature on this worldwide pest of coffee. I have some minor comments which should easily be addressed. I enjoyed reading this manuscript and look forward to seeing it published.

L87-88: How did you decide on the environmental conditions to maintain the lab colony (why 25 C and 80-96% RH)?

We reared the CBB at 25�C because when we were tried to establish the colony, we set the chamber at 27�C (as is used in most papers on growth the CBB on artificial diets), but the diets were drying and losing humidity too fast, so we set the chamber at 25�C. We put inside the chamber containers with distilled water to maintain the humidity of the diets. We didn’t use any salt solutions to control de humidity as is described in Portilla 2000. These conditions are working well to maintain our colonies. 

 Portilla, M. 2000. Development and evaluation of new artificial diet for mass rearing Hypothenemus hampei (Coleoptera: Scolytidae). Revista Colombiana de Entomología 26 (1-2): 31- 37. 

L124-125: Were entomological sleeves placed around the entire branch? If not, how do you know that artificial infestations were the result of lab-reared CBB and not naturally occurring CBB from neighboring trees/branches?

Yes, the plastic bags covered the entire branch. When we removed the bag from the branch, we observed some remaining dead and alive females in these bags. The maximum number of fruits perforated per branch was 15 to 17. We are sure that no CBBs entered the bags, and the artificial infestations in field were made by the CBB females reared in the lab and placed in the bags. 

L128: You might define larvae I vs. larvae II and pre-pupae vs. pupae.

We added three paragraphs in the introduction describing the different stages of the life cycle of the coffee berry borer. In these paragraphs we highlighted the differences between larva I vs. larva II, and prepupa vs pupa. See L 57 - 82. 

L135-138: Please explain why mortality by Bb and internal fruit rot was evaluated and how these variables relate to the overall goals of the study.

 We added a sentence in Materials and methods, explaining that natural infection with the fungus B. bassiana and fungi that cause internal fruit rot can kill CBB females. See L 195– 196. 

L141: “Fully feral” is the same as natural, correct? I think best to stick to “natural field-infestation” so as not to confuse the reader (it could be inferred that these were infestations in wild coffee). 

We agree with the reviewer. To avoid confusion, we changed ‘fully feral’ to ‘naturally.’ See L 184.

L1579-159: Are these daily temp and RH ranges, or ranges across the entire sampling period?

Yes, the values correspond to the period from August 11 to December 10 covering all our sampling period. These are the high and low values for the entire period. 

Table 1: Why is SE only shown for mean temp?

We added the SE for the other variables (relative humidity and rain), see Table 1. 

L172: Why were lab populations followed for 50 days and artificially infested populations followed for 56 days? Was this to account for the two days over which infestation could have occurred for artificial infestations?

The time for evaluations in the lab and field were different, because at 49 days in the field we didn’t observed the presence of juveniles or adults, while in the lab the first juveniles appeared at 28 days. we extended this time to 50 days to allow slower individuals (if any) to catch up, but we did not observe significant changes from day 45. 

L189: Here you say lab populations were evaluated at 56 days not 50 days?

In this line we report the time when the first juveniles and adults appeared in the field, not in the lab. Time of evaluations in the lab was 50 days and in field was 56 days. We revised and clarified this information throughout the manuscript. See L 214 and L 230 -231. 

L270-275: This should be moved to discussion.

We moved this information to discussion. See L 659 – 666. 

L327: How could both juveniles and adults show up on same day (day 56)? Wouldn’t the presence of adults on day 56 mean that juveniles had to be present before that? Does this mean you might have missed the first appearance of juveniles?

Yes, we agree with the reviewer. Probably our time between evaluations of 7 days was too long, some juveniles may have appeared during this period and were not counted immediately, but according to our results from lab evaluations the mean time from pupa to juvenile was 6.3 days. See Table 2

Table 2: Change “media” to “mean”. Please include SE for min and max.

We changed ‘media’ to ‘mean’. But we can’t include the SE for the values min and max, because in most of the cases we only have one data for each one. 

L523, L528: Change “parasite” to “parasitize”.

We made this change, See L 519. 

L526: Add “other” before predators.

We re-wrote this part of the Discussion, so this sentence was modified. 

L559-560: Yes, but this is because B. bassiana was not sprayed during your study, correct? Please include commonly reported rates of mortality due to B. bassiana from other studies.

We added two paragraphs about the range of natural infection observed with B. bassiana, and mortality observed for lab and field experiments done with local isolates and the strain GHA from the commercial products Mycotrol® and Botanigard®. See L526 – 538. 

L587: change to “more often recommended”.

We removed this part from the discussion, as reviewer #2 suggested we reduced this part. So this sentence was eliminated. 

L588: This is also true for other regions, such as Hawaii (see Johnson & Fortna et al. 2019).

We added information about the results of this study. See L 553 – 556. 

L602: add “adult”

We added the word ‘adult’ before ‘females’. See L 578. 

L648: Please explain why coffee species and variety would affect CBB development times.

After reviewing the literature, we decided to eliminate this phrase. We only left the effect of daily temperature on CBB developmental times that was our conclusion according with the literature reviewed and cited. See L 650 – 651. 

L691: I am confused by the statement “endosperm or almond from coffee fruits”. Please revise.

We deleted the word ‘almond’ to avoid confusion. See L 701. 

L704: Exposition? Do you mean exposure?

We change exposition to exposure, see L 714.

Reviewer #2

Abstract needs to be rewritten. Please provide some results. An abstract can not be based only on definitions and literature review.

We added two sentences to the Abstract incorporating values for demographic parameters and sensitivities observed for lab and field evaluations, see L 15 – 20 and L 25 -26. 

L 29 Add Coffea arabica L. (Gentianales: Rubiaceae)

We added this information, see L 33. 

L34 Add (Coleoptera: Curculionidae: Scolytinae)

We added this information, see L 39 -40.

L35 Add Coffee berry borer

We added this see L 41

L37-38 Add Citation to “only fertilized females abandon the fruit in search of new fruits to infest”

We cited the review from Vega et al. 2009, to support the statement that only fertilized females abandon the fruit in search of new fruits to infest, see L 45. 

Vega FE, Infante F, Castillo A, Jaramillo J (2009) The coffee berry borer, Hypothenemus hampei (Ferrari)(Coleoptera: Curculionidae): a short review, with recent findings and future research directions. Terrestrial Arthopod Reviews 2: 129-147.

L39 change the word adults by females

We change the word ‘adults’ to ‘females’, see L 46. 

L49 – Change Bb to B. bassiana (Do not use acronyms for scientific names)

 We change Bb to ‘B. bassiana’ in the entire document. 

L64 Include references from 18 and 21

We added other references to support this paragraph, now the references are from 21 – 25, the format for references required by the Journal put the references in that way. See L 101 

L76 Need citation (Portilla, 1999). "Portilla, M. 1999. Desarrollo y evaluation de una dieta artificial para la cria de Hypothenemus hampei. Rev. Cenicafe. 50:24-38"

We cited this reference, see L 114

L85 Change Coffee berry borers (CBBs) were with coffee berry borer was

We made this change, see L 124 

L 128 Delete the plural in CBBs and add the word stages

We made this change, see L 169. 

L 145 Here are you referring to the females? 

Yes, here we are referring to the females that is the most stage affected by the fungus B. bassiana. We changed ‘CBBs’ to ‘adult females’, see L 177. 

L146, L175, L223, L261, L265, L266, L269, L283, L284, L287, L332, L339 Delete the plural from CBBs

We deleted the plural from CBB all times that we are using this abbreviation to refer to the coffee berry borer, in some occasions we change the word CBB with females when we were refered to the females. We made these changes through the entire manuscript. 

L228 – 221 Focus on M&M. Leave definitions and literature review to introduction and discussion. Citation in M&M are used when procedures from somebody else's are used. 

We deleted all definitions from Materials and Methods. 

L221 Delete the abbreviation LTRE

We deleted the abbreviation from the start of the sentence, see L 261. 

L225-230 Same. Focus in M&M

We deleted all definitions from Materials and Methods. 

L271-275 Use these statements for discussion

We moved this information to discussion. See L 659 – 666. 

L286 What means Fate?

We changed ‘fate’ to ‘behavior’ to avoid confusion. See L 318

L300 Add ,CBB to the title of figure

To be consistent, for all titles of figures and tables we added coffee berry borer (CBB, Hypothenemus hampei)’ see L 318, 333, 348, 365, 378, 389, 405, 419, 438, 451, 461

L306 Change CBBs for coffee berry borer females

We made the change, see L338

L307 Add females after CBB

We made the change, see L339

L316 Add ,CBB to the title of figure and L332 Delete the plural from CBBs and add coffee berry borer before CBB

To be consistent, for all titles of figures and tables, we added ‘coffee berry borer (CBB, Hypothenemus hampei)’ see L 318, 333, 348, 365, 378, 389, 405, 419, 438, 451, 461

L339 Do not use plural. Check throughout the text

We deleted the plural from CBB all times that we are using this abbreviation to refer to the coffee berry borer, in some occasions we change the word CBB with females when we were refered to the females. We made these changes through the entire manuscript. 

L346 Add lab conditions: Temperature and Relative Humidity to the title of the table 2

We added this information, see L 379 - 380. 

L 360 -366 There is anyway to know if there is any significant differences between these cohorts? The rm values seem very low compared with other studies using Cenibroca artificial diet and parchment coffee under lab conditions:

1) Portilla et al.2000, (Reproductive potential response to continuous rearing of Hypotnenemus hampei developed using Cenibroca artificial diet. Rev. Colom. Entomol. 26:99-105.

2) Portilla and Street 2006 (Nuevas Tecnicas de production masiva automatizada de Hypothenemus hampei sobre la dieta artificial Cenibroca modificada. Rev. Cenicafe. 57:37-50) and 2008 (Profduccion masiva automatizada de la broca del cafe y de sus parasotoides sobre dietas artificiales. Sist. Agroeco. Mod. Biomatematic. 1:9-16)

3) Portilla et al. 2014 (Life tables as tools of evaluation and quality control for arthropod mass production, pp. 241-275. In: J. Morales-Ramos [Ed.], Mass Production of Beneficial Organisms. Academic Press, New York. 742 pp) (Book Chapter) look the case study "Abridged Life tables for the CBB, Hypothenemus hampei, at different temperatures using an artificial diet". 

All these articles can be cited as a reference and improved the current discussion.

We reviewed the references that the reviewer recommended. We added three paragraphs in the Discussion comparing the results from these studies with our results. We focused on the studies that reported demographic parameters for the CBB populations that were grown on the artificial diet Cenibroca. See L 590 – 616. 

L376 – 377 Keep the terminology uniform for all figures and tables if needed:

We made the titles more uniform using the phrase the “Coffee berry borer (CBB, Hypothenemus hampei)”

L450 Delete the sentence: Despite the importance of the CBB as pest of coffee and add It is well known the reproductive potential of the CBB on the Cenibroca artificial diet (Portilla, 1999, Portilla et al. 2000, Portilla and Streett 2008, and Portilla et al. 2014); however, only few studies have been conducted field experiments .....

 We made this change, see L 478 – 480. 

L466 Delete the heading: “Biological control as control for CBB management”

We made this change. 

L476 – 589 Go to the point. Your study is not related to natural enemies. This information can be reduced to one paragraph or two.

We agree with the reviewer, we reduced the information on natural enemies, for example, the section from ants as important predators of larvae and pupae of the CBB was reduced from five to two paragraphs. However, we consider that information about use of natural enemies to reduce the survival of larvae, pupae and adult females is important, and the information in the paper is novel in this context. We did not reduce this to one or two paragraphs as the reviewer suggested, but from sixteen to eight paragraphs. 

L 591 Delete the heading: “Life table parameters”

We deleted this heading. 

L592 I could not see in the table: the significant differences in life table parameters between populations reared in the lab vs. those from artificial infestations in the field. Please add that information.

The reviewer was right: we did not perform a statistical analysis for the life table parameters calculated for CBB populations reared in the lab vs. those from artificial infestations in field. We rewrote this sentence and changed ‘significant differences’ to ‘higher values in demographic parameters. See L 568 – 570. 

L614 This is one of the most complete study in demographic parameters using naturally infestation. So far, only two rearing systems have been able to get over that value Portilla 1999 in parchment coffee (0.073) and in Cenibroca diet (0.074) Portilla 2000. 

This information could be included in your discussion to support your conclusion

We added four paragraphs (see Lines 590 – 616) where we discussed only the studies which used the artificial diet Cenibroca, since they are the most directly comparable. We think that the studies where CBB was grown on parchment coffee are less relevant, and would increase the length of the Discussion. 

L 620 did you mean differed?

No, we are using ‘similar’ to say that these values are almost equal, not different. 

L623 Delete the plural from CBBs

We made this change in the entire manuscript. 

L624 include demographic data obtained in Cenbroca diet by other authors.

We added this information, see Lines 590 - 616

L642 Definitely, temperature affect demographic values. Please see "Abridged Life tables for the CBB, Hypothenemus hampei, at different temperatures using an artificial diet"

We added this information, see Lines 598 – 603

L 650 Delete the heading: “Comparissons between CBBs rearing in the lab vs. CBBs growing on the field”

We deleted this heading. 

L 726 – 729 Based in this study, it is important to know the real effect of the natural enemies on the demographic values of the CBB. 

Caged and uncaged coffee trees should be the answer.

We added this sentence: “to determine the impact of natural enemies on the demographic parameters of the CBB” at the end of the Conclusion. See L 740 – 743.

---

## [Decision Letter · Decision Letter 1]

25 Oct 2021

PONE-D-21-18910R1Demography and perturbation analyses of the coffee berry borer *Hypothenemus hampei* (Coleoptera: Curculionidae): implications for managementPLOS ONE

Dear Dr. Mariño,

Thank you for submitting your manuscript to PLOS ONE. After careful consideration, we feel that it has merit but does not fully meet PLOS ONE’s publication criteria as it currently stands. Therefore, we invite you to submit a revised version of the manuscript that addresses the points raised during the review process.

We look forward to receiving your revised manuscript.

Kind regards,

Patrizia Falabella

Academic Editor

PLOS ONE

Journal Requirements:

Reviewers' comments:

Reviewer's Responses to Questions

**Comments to the Author**

1. If the authors have adequately addressed your comments raised in a previous round of review and you feel that this manuscript is now acceptable for publication, you may indicate that here to bypass the “Comments to the Author” section, enter your conflict of interest statement in the “Confidential to Editor” section, and submit your "Accept" recommendation.

Reviewer #1: All comments have been addressed

Reviewer #2: (No Response)

2. Is the manuscript technically sound, and do the data support the conclusions?

Reviewer #1: Yes

Reviewer #2: Yes

3. Has the statistical analysis been performed appropriately and rigorously? 

Reviewer #1: Yes

Reviewer #2: Yes

4. Have the authors made all data underlying the findings in their manuscript fully available?

Reviewer #1: Yes

Reviewer #2: Yes

5. Is the manuscript presented in an intelligible fashion and written in standard English?

Reviewer #1: Yes

Reviewer #2: Yes

6. Review Comments to the Author

Reviewer #1: Methods, results, objectives were correctly analyzed and interpreted. Authors addressed all comments and suggestions.

Reviewer #2: Review 2: Marino et al.

L9: change to “entire life cycle occurs inside the fruit where it is well-protected.”

L10: change to “would shed light on the population dynamics of this pest…”

L20: change comma to semi-colon

L21-22: change to “indicated that effective CBB management should consider multiple developmental stages;…”

L48: change to “which cause internal fruit rot”

L50: change to “because chemical and biological control measures are only effective during the short period of time when adult females are out searching for new fruits to infest.”

L164-166: You need to include here that entomological sleeves were used to cover the experimental branches.

L177: note in parentheses at first mention that Bb refers to B. bassiana

L194-196: Move this up to start the paragraph at L177.

L214: You need to include the information in the text here on why you used different evaluation periods for lab and field (50 vs. 56 days). Please include the information you put in your response to reviewers into the manuscript here.

L530: indicating that?? farmers should use commercial formulations to manage CBB instead of relying on naturally occurring B. bassiana strains? This thought seems incomplete as written.

L536: Why do you think the mortality by natural Bb was higher in the study reported by Wraight et al? Are the farms in PR considerably drier and hotter than farms in Hawaii where feral strains of Bb thrive?

L552, 555, 559: change sanitization to sanitation

L590-603: Here you should include the information from your response on why you used 25 C instead of 27 C.

7. PLOS authors have the option to publish the peer review history of their article (what does this mean?). If published, this will include your full peer review and any attached files.

Reviewer #1: No

Reviewer #2: No

---

## [Author Response · Author response to Decision Letter 1]

8 Nov 2021

Dear Dr. Chenette:

We have revised the manuscript PONE-D-21-18910R1following the suggestions of the one anonymous reviewer, also we carefully revised the references to adjust our manuscript for publication in PLOS ONE. We found the comments to be very useful and constructive, and we have incorporated most of them. The list of comments and our responses (in bold) follow below.

In this revised version, we followed the suggestions made by the reviewer #2. We added two new paragraphs, one in the methods section and other in the discussion. The first was added to explain the difference in the time for evaluations between field and lab. The second was added to explain the reasons to growth the insects or CBB at 25 �C instead to 27 �C, that is the optimal temperature recommended to growth the CBB in artificial diets. 

The new paragraphs and/or modifications in the manuscript that we made in this revision are shown in red text. 

We appreciate the opportunity to resubmit this revised version of our manuscript and look forward to hearing from you. If we can be of any assistance, please do not hesitate to contact us. 

Sincerely,

Yobana Andrea Mariño Cardenas

Department of Biology

University of Puerto Rico - Rio Piedras San Juan, PR 00931-3360

Reviewer #2. 

L9: change to “entire life cycle occurs inside the fruit where it is well-protected.”

We made the change, see L9. 

L10: change to “would shed light on the population dynamics of this pest…”

We made the change, see L10 -11.

L20: change comma to semi-colon

We made the change, see L20.

L21-22: change to “indicated that effective CBB management should consider multiple developmental stages;…”

We made the change, see L22.

L48: change to “which cause internal fruit rot”

We made the change, see L48.

L50: change to “because chemical and biological control measures are only effective 

during the short period of time when adult females are out searching for new fruits to infest.”

We made the change, see L50-51.

L164-166: You need to include here that entomological sleeves were used to cover the experimental branches.

We used plastic bags to cover the entire branch. We added a phrase to specify that plastic bags were used as entomological sleeves, see L 163-164. 

L177: note in parentheses at first mention that Bb refers to B. bassiana

To be consistent we changed Bb to B. bassiana, as you recommended in the first revision, see L 181. 

L194-196: Move this up to start the paragraph at L177.

We made the change, see L 177-179. 

L214: You need to include the information in the text here on why you used different evaluation periods for lab and field (50 vs. 56 days). Please include the information you put in your response to reviewers into the manuscript here.

We added the information, see L 212 -216. 

L530: indicating that?? farmers should use commercial formulations to manage CBB instead of relying on naturally occurring B. bassiana strains? This thought seems incomplete as written.

We added a sentence recommending the application of both commercial and local strains of B. bassiana to reduce the damage caused by the coffee berry borer, see L 532 – 533. 

L536: Why do you think the mortality by natural Bb was higher in the study reported by Wraight et al? Are the farms in PR considerably drier and hotter than farms in Hawaii where feral strains of Bb thrive?

There are two possible explanations for the difference in the natural presence or mortality of the CBB caused by B. bassiana between Hawaii and Puerto Rico. The first, we agree with you, the differences in temperature and relative humidity could be affecting the presence and development of natural infections with Bb. 

Farms from Puerto Rico experienced hotter and dryer days; for example, in our sampling site at 434 m.a.s.l. the temperatures ranged from 13 to 34�C and the minimum relative humidity was 47.6%; these measures are from September to November. While in Hawaii, for this same interval of time in a site located at a similar altitude 427 m.a.s.l, the temperatures ranged from 18.9 to 29.8�C and the minimum relative humidity was 66.7%. 

Another reason could be that our sampling site was located at an experimental farm with similar management to a commercial Farm. Wraight et al, 2018 found that the natural presence of Bb was significantly less common in commercial farms than feral sites. 

Environmental data from Hawaii were taken from Hamilton et al. 2019. Coffee berry borer (Hypothenemus hampei) (Coleoptera: Curculionidae) development across an elevational gradient on Hawai‘I Island: Applying laboratory degree-day predictions to natural field populations. PLOs ONE 14(7): e0218321. https://

doi.org/10.1371/journal.pone.0218321.

Wraight et al. 2018. Prevalence of naturally-occurring strains of Beauveria bassiana in populations of coffee berry borer Hypothenemus hampei on Hawai'i Island, with observations on coffee plant-H. hampei-B. bassiana interactions. J. Invertebr. Pathol. 156, 54–72. doi: 10.1016/j.jip.2018.07.008.

L552, 555, 559: change sanitization to sanitation

We made the change, see L 555, 558, 562. 

L590-603: Here you should include the information from your response on why you used 25 �C instead of 27 �C.

We added the information, see L 606-608.

---

## [Editor Report · Decision Letter 2]

11 Nov 2021

Demography and perturbation analyses of the coffee berry borer *Hypothenemus hampei* (Coleoptera: Curculionidae): implications for management

PONE-D-21-18910R2

Dear Dr. Marino,

We’re pleased to inform you that your manuscript has been judged scientifically suitable for publication and will be formally accepted for publication once it meets all outstanding technical requirements.

Kind regards,

Patrizia Falabella

Academic Editor

PLOS ONE

---

## [Editor Report · Acceptance letter]

2 Dec 2021

PONE-D-21-18910R2 

Demography and perturbation analyses of the coffee berry borer *Hypothenemus hampei * (Coleoptera: Curculionidae): implications for management 

Dear Dr. Mariño:

I'm pleased to inform you that your manuscript has been deemed suitable for publication in PLOS ONE. Congratulations! Your manuscript is now with our production department. 

Kind regards, 

on behalf of

Prof. Patrizia Falabella 

Academic Editor

PLOS ONE